# Group 2 Innate Lymphoid Cells: A Double-Edged Sword in Cancer?

**DOI:** 10.3390/cancers12113452

**Published:** 2020-11-20

**Authors:** Enrico Maggi, Irene Veneziani, Lorenzo Moretta, Lorenzo Cosmi, Francesco Annunziato

**Affiliations:** 1Immunology Department, Bambino Gesù Children Hospital, 00165 Rome, Italy; irene.veneziani@opbg.net (I.V.); lorenzo.moretta@opbg.net (L.M.); 2Department of Experimental and Clinical Medicine, University of Florence, 50134 Florence, Italy; lorenzo.cosmi@unifi.it (L.C.); francesco.annunziato@unifi.it (F.A.)

**Keywords:** group 2 innate lymphoid cells, immunity in tumors, immunotherapy

## Abstract

**Simple Summary:**

Group 2 innate lymphoid cells (ILC2s), like other ILCs, are a new resident cell subset of innate immunity that provides the first line of defense against pathogens such as helminths and largely contributes to inflammation observed in allergic disorders and in some tumors. They are considered sentinel cells, which reside at the interface between host and external environment and are rapidly activated by signals deriving from tissue. Depending on the type of signals and their high plasticity, ILC2s exert both enhancing and regulatory activity on other tissue-resident cells, including tumor- and tumor-associated cells. The functional profile of ILC2s, their pro- or antitumor activity in preclinical studies and patients and the potential therapeutic approaches targeting ILC2s have been extensively reviewed.

**Abstract:**

Group 2 Innate Lymphoid Cells (ILC2s) belong to the family of helper ILCs which provide host defense against infectious agents, participate in inflammatory responses and mediate lymphoid organogenesis and tissue repair, mainly at the skin and mucosal level. Based on their transcriptional, phenotypic and functional profile, ILC2s mirror the features of the adaptive CD4+ Th2 cell subset, both contributing to the so-called type 2 immune response. Similar to other ILCs, ILC2s are rapidly activated by signals deriving from tissue and/or other tissue-resident immune cells. The biologic activity of ILCs needs to be tightly regulated in order to prevent them from contributing to severe inflammation and damage in several organs. Indeed, ILC2s display both enhancing and regulatory roles in several pathophysiological conditions, including tumors. In this review, we summarize the actual knowledge about ILC2s ability to induce or impair a protective immune response, their pro- or antitumor activity in murine models, human (children and adults) pathologies and the potential strategies to improve cancer immunotherapy by exploiting the features of ILC2s.

## 1. Introduction

Helper Innate Lymphoid cells (ILCs) are classified into four groups (ILC1s, ILC2s, ILC3s and lymphoid tissue-inducer—Lti—cells) mimicking the functional profiles of adaptive CD4+ Th cell subsets. ILC2s are devoted to defending against pathogens, lymphoid organogenesis, tissue repair and type 2 inflammatory response in many immune-mediated disorders, including cancer [1].

Exploiting their multiple receptors, ILC2s can be rapidly activated by signals derived from tissue, thus playing a role as sentinel cells involved in the first line of defense against pathogens [2,3,4,5,6]. The biological activities of ILCs need to be regulated to not improve severe inflammation and damage in several organs [7]. Indeed, ILC2s exert both enhancing and homeostatic activities on several cells, including cancer cells and cells from tumor microenvironments (TMEs). Herein we will summarize the main features of ILC2s, analyze their pro- or antitumor activity in different tumors and discuss the ILC2-targeted strategies to improve cancer immunotherapy.

## 2. Main Features of ILC2s

ILCs derive from common lymphoid progenitors (CLPs) from which common innate lymphoid progenitors (CILPs) and common helper innate lymphoid progenitors (CHILPs) are generated. Whereas CILPs differentiate into natural killer (NK) cell progenitors (NKPs) which then generate NK cells, CHILPs divide into innate lymphoid cell progenitors (ILCPs) and lymphoid tissue inducer progenitors (LTiPs) [1,8,9,10,11]. Finally, ILCPs generate helper-like ILC subsets, including ILC2s. The transcriptional repressor Id2 is required and sequentially expressed by ILCPs precursors, which can further differentiate, with lineage-specific transcription factors mirroring the phenotype and the function of adaptive helper T cells subsets [8].

ILC2s mainly reside in the submucosa of lung and gut as well as in derma and fat tissue. In these strategic sites they behave as initiators of type 2 immune response. Importantly, ILC2s also localize at interfollicular regions surrounding B cell follicles at the entry of the afferent lymphatics of lymph nodes where interactions between T and B cells occur. This suggests that ILC2s can influence such interactions or the beginning of humoral immune responses by also exploiting their established antigen presenting cells (APC) function [12].

Tissue-resident ILC2s display a panel of sensors for inflammatory mediators, pathogen- and damage-associated molecular pattern (PAMPS and DAMPS), cell density, neuronal signals, complement and angiogenic factors. They are mainly activated by epithelial cells that secrete inflammatory mediators in response to pathogens or other environmental stimuli, thus starting the local immune response [12,13,14]. Epigenetic regulatory networks allow rapid expression of ILC2s functions in response to microenvironmental cues, thus enabling these sentinel cells to promptly respond by restoring homeostasis and collaborating with other resident cells (tissue-resident T memory cells, unconventional T cells, etc) to activate local immunity and recruit different type of circulating cells [1]. As for Th2 cells, human ILC2s are a GATA-3-dependent subset, even though other transcription factors (RORα, GFI-1, TCF1, Runx1 and Bcl11b) are essential for their development and function [15,16,17,18,19,20,21].

ILC2s do not express rearranged antigen receptors, but integrate multiple (mainly soluble) signals by expressing several receptors that mediate different functions [22]. Their phenotype is characterized by the expression of the prostaglandin D2 (PGD2) receptor 2 (CRTH2), the IL-33 receptor (ST2) and variable levels of c-Kit [23], as well as the NK cell receptor CD161 [24] and the NCR NKp30 [25], which, when engaged by B7-H6, leads to the activation of skin-derived ILC2s. Moreover, the killer cell lectin-like receptor subfamily G member 1 (KLRG1, a coinhibitory receptor expressed by T and NK cells binding the E-cadherins) arises during ILC2 development [26,27]. Notably, ST2+KLRG1+/− ILC2s, defined as natural ILC2s (nILC2s), are responsive exclusively to IL-33, while ST2− KLRG1^hi^ ILC2s, called inflammatory ILC2s (iILC2s), differentiate during infections. iILC2s are highly responsive to IL-25 [28] and can differentiate/shift into ILC3-like cells under type 3-promoting molecules stimulation [26]. Recently it has been shown that helminth-induced IL-33 promotes the generation of iILC2s through the expression of tryptophan hydroxylase 1 and ICOS [29].

ILC2s are triggered by a wide range of soluble mediators released by different types of hematopoietic or nonhematopoietic cells, (epithelial-, adipose-, mast- and tumor cells) [21,30,31,32,33,34], through many activating and inhibitory receptors, whose knowledge is crucial to understanding their physiologic and pathophysiologic activity [22].

The main ILC2-stimulating signals are alarmins (IL-25, IL-33 and thymic stromal lymphopoietin—(TSLP-)), growing cytokines (IL-2, IL-4, IL-9 and IL-7) and eicosanoids (mainly PGD2). Besides alarmins receptors, IL-2Rα- (CD25) and IL-7R are essential for the development, homeostasis and activation of ILC2s. ILC2s can also be stimulated by neuropeptides, such as neuromedin U (NMU) and vasoactive intestinal peptide (VIP), which contribute to cytokine production (mainly IL-5) for the eosinophil homeostasis in the lung and gut [35,36]. ICOS and its ligand ICOSL are coexpressed on ILC2s, and their interaction promotes the proliferation of ILC2s as a self-amplifying mechanism [37]. Among activating receptors, ILC2s express several Toll-like receptors (TLRs) (TLR1, 2, 4, 6): TLR2 can directly activate lung ILC2s, an effect potentiated by some allergens. We have shown that human-circulating ILC2s exhibit higher TLRs expression than autologous Th2 cells, and their stimulation through specific ligands induces IL-5 and IL-13 production [38]. Finally, ICAM-1 and its ligand LFA1 are highly expressed on ILC2s and their progenitors, and IL-33 enhances their expression. In particular, ICAM-1 deficiency impairs ILC2s development and function, contributing to reduce lung allergic inflammation [39].

Among the inhibitory receptors, the calcitonin gene-related peptide (CGRP) negatively modulates ILC2s functions in lung inflammation or during helminth infections [40,41]. The β2-adrenergic receptor, highly expressed by intestinal ILC2s, also acts as a negative regulator of the ILC2-mediated inflammatory response [42]. Other inhibitory receptors are those of IL-27, type I or type II IFNs, PGE2, or regulatory cytokines produced by inducible T regulatory cells (Tregs), each of them controlling ILC2s functions through different mechanisms [43,44].

When activated, ILC2s secrete type 2 cytokines, such as IL-4, IL-5, IL-9, IL-13 and amphiregulin (AREG, a member of the epidermal growth factor (EGF) family), which are involved in tissue repair, airway responses and helminth worms expulsion [45]. AREG enhances the suppressive activity of Tregs by triggering surface EGF receptors (EGFR), tissue repair and upregulating the TGF-β production [45]. Activated ILC2s produce PGD2 that acts in an autocrine way supporting ILC2s function via the receptor CRTH2 [46]. In contrast to murine ILC2s, we have recently shown that human-circulating ILC2s from healthy donors produce IL-4, and that the frequency of IL-4-producing ILC2s is higher in atopic individuals than in healthy donors [39].

ILCs are highly plastic and have the potential to transdifferentiate from one subset into another in response to environmental cues [47]. Whereas IL-4 is crucial for retaining ILC2s identity, in respiratory-related disorders, human circulating ILC2s are shown to shift into functional ILC1-like cells that secrete IFN-γ in response to IL-12 and IL-1β [48,49,50]. ILC2s/ILC1s conversion requires the induction of T-bet and the presence of IL-12R. It isn’t known if the shifted ILC1s-modulated cells acquire cytolytic activity particularly in TME or derive from a cytotoxic ILC2s subset coexpressing CRTH2 and CD94, a phenotype rarely observed in normal tissues. However, since CRTH2 is downregulated in ILC2s upon activation, it is likely that cytotoxic CRTH2-negative ILC2s are missed in ex vivo analysis of TME infiltrating cells [51]. Moreover, the exposure of murine ILC2s to Notch ligands-induced RORγt expression and elicited both IL-17 and IL-13 production [52]. In humans a c-kit+CCR6+ILC2s subset can shift into IL-17-producing NKp44− ILC3-like cells expressing RORγt in response to IL-1β and IL-23 [53]. We have recently shown that IL1β and IL-23 can favor human circulating ILC2s shift to express RORC and secrete IL-22 [50]. In vitro, the modulated ILC2s continue to express GATA-3 and type 2 cytokines, which are only partially reduced. Of note, such modulated ILC2s have reduced ability to drive IgE synthesis by autologous B cells, since their expression of CD154 (CD40L) is downregulated [38,50], whereas the helper activity for IgG, IgM and IgA remains unaltered.

## 3. Enhancing and Regulatory Function of ILC2s

Preclinical studies have clearly shown that ILC2s promoted Th2-cell differentiation since, in their absence, Th2 response in mice is impaired when challenged with allergens or parasitic worms [15,54]. Accordingly, in humans, an increased number of ILC2s and activity were observed in patients with Th2-oriented disorders as respiratory allergy, atopic dermatitis, chronic rhinosinusitis and eosinophilic esophagitis [45,54,55,56,57,58,59,60]. Notably, intrahepatic ILC2s contribute also to the process of fibrogenesis in liver diseases through secretion of AREG [61].

ILC2s are the main producers of IL-5, which play a key role in eosinophilic inflammation promoting the differentiation of eosinophils in the bone marrow, their recruitment and survival in the lung of patients with respiratory allergy. When activated the eosinophil produce toxic mediators, that are crucial in determining tissue damage and remodeling [36]. Among such mediators, EDN has been recognized to stimulate TSLP from myeloid dendritic cells (mDC), which in turn maintains this type of inflammation by increasing the ILC2s’ survival [62]. ILC2s have been associated in humans with more vigorous variants of airway type 2 inflammation that occur in older asthmatic patients characterized by high numbers of airway and blood eosinophils but relatively low concentrations of serum IgE [45,63].

The capacity to produce IL-4 and IL-13 has a greater impact, allowing ILC2s to interact with cells of both adaptive and innate immunity. ILC2s support humoral immunity which includes the proliferation/activation of T cell-independent innate B1-cell response and T cell-dependent acquired B2-cell response in mice [47]. In humans, we have recently shown that activated ILC2s express CD154 and are able to induce in vitro IgE production by autologous B cells [38]. This may explain why allergen-specific IgE is often associated to polyclonal IgE production in allergic disorders. Since ILC2s localize at interfollicular regions of lymph nodes where T-B interactions occur, they can directly stimulate B cells to produce polyclonal IgE, but also may influence specific Th2-B cell cooperation for starting allergen-specific IgE synthesis, exploiting their APC function. Indeed, it has been shown that ILC2s may exert APC activity for naïve CD4+ T cells since they express MHC-Class II and costimulatory molecule OX40L [45,58,64]. IL-4 has also trophic activity on basophils and mast cells (MC) which are short-lived circulating cells recruited to inflammatory sites (basophils) or long-lived tissue resident cells (MC) [45,47,55,59].

Finally, other molecules, such as IL-13 and AREG from activated ILC2s, collectively elicit mucus production, possess smooth muscle hypercontractility, and possess tissue repair abilities, improving fibrogenesis [55,65,66].

Concerning their regulatory activity, ILC2s can promote Tregs expansion and downregulate excessive immune responses. Tregs proliferation is induced by direct contact with ILC2s through OX40/OX40L or ICOSL/ ICOS signaling [65,67]. A subset of IL-9-producing ILC2s expressing ICOSL and GITRL, plays a role by activating Tregs in murine and human arthropathies [68]. An increased proportion of IL-9+ILC2s was observed in the blood and joints of patients in remission with rheumatoid arthritis compared to those with relapsed disease: this suggests that IL-9+ ILC2s are essential for the resolution of process of inflammation likely due to other regulatory cytokines that these cells produce [45,69].

Another regulatory subset is IL10-producing ILC2s. IL-10 exerts suppressive effects on DC, macrophages and Th2 cells, and stimulates the expansion of Tregs [70]. Interestingly, these cells shift from the fatty acid oxidation pathway conventionally used for proinflammatory effector functions, to the glycolytic pathway for IL-10 production [71]. Few IL-10+ ILC2s have been detected in the intestine [47] where their expression was induced in vitro by IL-2, IL-4, IL-27, IL-10 and NMU. Notably this may link the IL-4 stimulation with the suppressive activity of ILC2s. Secreted IL-10 further increased IL-10 production by ILC2s through a positive feedback loop, while TNF superfamily member TL1A strongly inhibited IL-10 production of intestinal ILC2s [72].

In conclusion, by balancing their inhibitory and enhancing activity, ILC2s may orchestrate the protective type 2 response and tissue repair as well as the onset and the maintenance of allergic inflammation.

## 4. Pro- and Antitumor Activity of ILC2s

A high number of ILC2s have been detected mainly in IL-33-enriched tumors, such as breast, gastric and prostate cancer [73,74], since IL-33 is the main ILC2s activator and usually promotes tumor growth, metastases and angiogenesis [75]. An increased proportion of ILC2s have been also found in pancreatic adenocarcinoma where tumor cells and tumor cell-conditioned macrophages produce IL-1α and IL-1β favoring TSLP secretion by cancer-associated fibroblasts [76]. Several primary tumors and tumor cell lines have been described to produce alarmins as IL-25 and TSLP [77,78]. However, as described, IL-33 play the major role in ILC2s activation since IL-25 activate only iILC2s and TLSP may trigger TSLP receptors expressed by tumor cells in an autocrine manner and independently from a type 2 inflammation. Considering enhancing and regulating properties of ILC2s, their role in TME is still controversial. Moreover, ILC2s activity in tumors could depend on ILC2s distribution and differentiation in tissues, and molecules release by tumor in TME.

Herein we will summarize the mechanisms induced by activated ILC2s favoring or opposing tumor growth and its diffusion [32,79,80,81,82,83,84,85,86,87] (Table 1).

### 4.1. Protumor Activity

The protumor activity of ILC2s is mainly attributed to IL-33-driven IL-4 and IL-13 production from these cells which have been reported to support tumor development and progression [88]. Depending on the TME and the study models, ILC2s may favor tumor escape through several complementary mechanisms: i. recruitment/activation of myeloid-derived suppressor cells (MDSCs) and/or Tregs which inhibit antitumor CD8+T cell response [32,75,84], ii. impairment of NK cell-mediated tumor killing, iii. development of type 2 response conditioning TME and tumor cells, iv. induction of epithelial cell proliferation/transformation.

*Establishment of ILC2–MDSC regulatory axis.* In a breast cancer model, it has been shown that IL-33 treatment increased tumor growth and metastases [75]. Mice from this model displayed increased proportion of ILC2s and MDSCs (CD11b+CD11c+Gr1+Ly6G−Ly6C+) in the spleen and TME, as well as upregulated IL-13 serum level. Accordingly, ST2 (IL33R) KO mice had reduced levels of MDSC [75]. In this model, IL-13 derived from tumor-activated ILC2s was suggested to directly induce MDSC with suppressive activity on antitumor T cell response, as confirmed by MDSC reduction upon partial ILC2 depletion. Recently, the ILC2-MDSC immune-regulatory axis has been established in human bladder and prostate cancers, as well as in acute promyelocytic leukemia (APL) [32,86]. It has been shown that the protumor function in APL is mediated by high levels of PGD2 and B7-H6, able to expand and activate ILC2s [32]. The over-production of IL 13 by ILC2s and MDSCs, in turn, induced a strong immunosuppression mainly of antitumor adaptive immunity. Additional evidence of this suppressive axis includes: i. the proportions of ILC2s and MDSC in the urine of patients with bladder cancer and receiving intravesical Bacillus Calmette–Guerin (BCG) therapy were negatively correlated with patients’ outcome, which was also the case in murine models of prostate tumor; ii. ILC2s in prostate and bladder murine cancer secrete IL-13, whose receptor (IL-13Rα1) is highly expressed on monocytes and MDSCs; iii. IL-13 triggering induces markers of suppressive function in monocytic cells (Arg1, iNOS, C/EBPβ, IL1-RA). In human gastric and lung tumors a correlation between circulating ILC2 and MDSC were also observed, suggesting an active ILC2-MDSC suppressive axis in several tumor isotypes [81,86,88].

*Increase of Tregs.* Some reports indicate that ILC2s favor the Tregs compartment, which impairs antitumor T cell responses and is usually associated with poor prognosis [89]. AREG, produced by ILC2s, induces Tregs through TGF-β production [45,90]. ILC2s also expand Tregs via an OX40L and ICOSL-dependent mechanisms in murine models of allergen exposure and helminth infection [64,67]. IL-33-treated tumor-bearing mice also showed an elevated proportion of CD4+FoxP3+ Tregs [90], due to the direct effect of IL-33 on Tregs expressing ST2. In addition to the above-mentioned OX40L-dependent ILC2-mediated Tregs expansion, even the increased MDSCs may in turn attract and induce Tregs proliferation [91,92]. On the whole, ILC2s may directly or indirectly favor the involvement of Tregs, even though this must be formally shown inside TME. Conversely, ILC2s themselves can be the target of regulatory cells or cytokines: for instance we have recently shown that the proliferation, cytokines production and CD154 expression of human ILC2s are inhibited by CD4+CD25high Foxp3+ Tregs, while TGF-β reduces CD154 expression on ILC2 stimulated with IL-25/IL-33 [38]. A subset of hyporesponsive IL-10-producing ILC2s expressing TIGIT and PD1 were found in the lung of allergic severe inflammation [69]. Transcriptome analysis revealed similarities of this subset with exhausted CD8+ T cells observed during chronic viral infections; for this reason they have been designed “exhausted-like” ILC2s [47,69]. Thus, it cannot be excluded that immunosuppressive TME, as other chronic inflammatory tissue, may promote the proliferation of ILC2s subsets with regulatory activity at least in some kind of tumors [68,70]. However, the role of the previously described IL10+ILC2s in tumor progression as well as the signals promoting their generation/recruitment in TME, are still unknown.

*Inhibition of NK cells.* As we will report later, IL-33 was also shown to inhibit tumor growth in murine models by improving antitumor innate and adaptive immunity [92,93,94]. Notably, NK cell activity on melanoma tumor growth was improved, when IL-33-activated ILC2s were depleted [94], because they strongly upregulated CD73 ecto-enzyme which, in vivo, suppress T and NK cell functions by upregulating the suppressive adenosine in TME. [95,96,97]. CD73−/− KO ILC2s were not able to suppress NK cell activity in vitro compared to normal ILC2s from wild-type mice [94], suggesting that they suppress NK cells via CD73-mediated activity. On the other hand, NK cells downregulate ILC2s since their depletion in murine lung inflammation increased ILC2 levels and ILC2s-mediated cytokines [97].

*Development of Type 2-oriented microenvironment*. In humans, tumors with type 2-polarized inflammation is usually associated with poor prognosis, due to the ability of Th2 and M2 cells to modulate effectors cells far from a protective antitumor (type-1)-immune response [55,98]. Similarly, ILC2s may show a harmful effect by blocking type-1-immunity, potentiating Th2 cells expansion. In this context, IL33-induced OX40L on ILC2s seems to be crucial for their ability to induce both Th2 cells and Tregs [64]. Thus, a vicious circle amplifying protumor response is established between ILC2s and Th2 cells which potentiate each other through their cytokines [99].

IL-4 and IL-13 per se exert a protumor function, since they directly bind to high-affinity receptors (IL-4Rα and IL-13Rα1 chains), forming functional receptors in cancer cells [100]. IL-13 also binds with the high affinity IL-13Rα2 private chain, leading to the induction of TGF-β with protumor activity. Accordingly, the IL-13Rα2 expression by tumor cells correlates with metastases in multiple cancers [101,102]. The IL-4 and IL-13 signaling mediate biological effects, such as tumor proliferation, cell survival, cell adhesion and metastasis. In certain cancers, the presence of these cytokine receptors may serve as biomarkers of cancer aggressiveness [88]. IL-4 and IL-13 may contribute to type 2-oriented TME by stimulating mDCs to produce Th2-attracting chemokines (CCL17) [103] and promoting M2-type macrophages further favoring both Th2 differentiation and immunosuppressive environment [104]. IL-13 is also able to stimulate mDC expressing FcεR1 to migrate into the lymph nodes and trigger Th2 response. Finally, ILC2s recruit other cells such as eosinophils and mast cells (MC), which may in turn potentiate a type 2-oriented environment with protumor activity.

Eosinophils recruited by ILC2s-derived IL-5 is an important driver of mDC-mediated Th2 cell differentiation. When activated these cells produce toxic mediators, that are crucial in determining the damage and remodeling of epithelia which overproduce ILC2s activating alarmins (IL-33 and TSLP) [36,105]. Interestingly, monocyte-derived mDC themselves can produce TSLP upon stimulation by microbial products, suggesting that it may induce type 2 inflammation in an autocrine manner [106]. In addition, the eosinophil mediator EDN has been also recognized to stimulate TSLP from mDC, which, in turn, amplifies this type of inflammation by increasing the ILC2s survival in an autocrine fashion [62]. ILC2s-driven eosinophils promote tumor growth by other mechanisms [107]: i. the production of CCL22 which facilitate the migration of Tregs into tumors, ii. the expression of indoleamine 2,3-dioxygenase (IDO), which inhibits effector T cells, iii. the production of EGF and TGF-β1 inducing tumor growth and epithelial mesenchymal transition, respectively, iv. the neosynthesis of IL-4 and IL-13 (promoted by TSLP) which polarizes macrophages to M2 profile, v. the production of metalloproteinases (MM2 and MM9) inducing matrix remodeling and facilitating metastases.

Finally, it has been suggested that MC may promote small bowel cancer in mice. It has been shown that, initially, IL-10+ regulatory T cells may favor MC expansion and polypus growth while, later on, a second MC subset expands during the transition from polypus to carcinoma [108]. In the TME of this model, ILC2s were increased compared with the healthy surrounding tissues, as indicated by the high expression of IL-5 and IL-13 [108].

*PD1 expression on ILC2s.* While the expression of some immune checkpoints such as TIM3, TIGIT, LAG3 are only partially known, the PD1 expression on ILC2s is well-documented [109]. PD1 is detectable on ILCp, lost during differentiation and upregulated on tissue ILC2s upon inflammation, mainly in the lung [110]. In humans, PD1 expression has been described on ILC2s, but not ILC1s or ILC3s. It acts as a negative regulator of mature KLRG1+ ILC2s function depending on type of inflammation, the tissue origin and the strength of IL-33 stimulation, leading to reduced ILC2s function. The PD1L/PD1 interaction limits the proliferation and cytokine secretion of mature ILC2s by inhibiting the phosphorylation of STAT5 [111,112]. Besides cell functions, PD1 also acts as a metabolic checkpoint on ILC2s, affecting glycolysis, glutaminolysis and methionine catabolism. In agreement with its regulatory activity, PD1 triggering ameliorates airway hyperreactivity and suppresses ILC2s-driven lung inflammation in a murine asthma model [113]. PD1 and CTLA-4 are also coexpressed by ILC2s infiltrating TME of some tumors as breast cancer. In gastrointestinal tumors a higher expression of PD1 in malignant- compared to para-lesioned tissues was shown for ILC2s [74]. Even though the role of PD1 expression on ILC2s in tumors remains largely unknown, however, the result that it is possible to potentiate ILC2s function by using PD1-blocking antibodies suggests that this therapy can also affect type 2 responses that usually favors cancer growth. However, this is not true for all types of cancer: as reported later, it has been recently shown that ILC2s infiltrating pancreatic ductal adenocarcinomas (PDAC) are able to activate tissue-specific tumor immunity [87]. In this tumor both resting and activated ILC2s express PD1 and they further expand with PD1 blockade (αPD-1) to enhance tumor control by tumor-specific CD8+ T cells. In agreement, in these patients, the higher proportion of tumor ILC2s correlated with longer survival [87]. At present, far too little attention has been paid to ILC2s-mediated immunomodulation in tumors by PD1 and PD1L interaction and more detailed studies are required to investigate such relationship.

*Proliferation/transition of epithelia.* Among the various activities of ILC2s, tissue repair is crucial in maintaining airway epithelial integrity and remodeling during viral infections. This activity is prevalent due to molecules such as AREG, IL-4 and IL-13 which induce fibrogenesis by directly acting on fibroblast receptors or through profibrogenic cytokines (TGF-β) [45,47]. The carcinogenesis-promoting role of IL-33/ILC2s/IL-13 axis on epithelial cells has been suggested, since: i. the stimulation of this axis promotes liver fibrosis [114], ii. IL-33 is elevated in sera of patients with biliary atresia [115], iii. IL-33 triggers ILC2/IL-13 mediated proliferation of cholangiocytes and epithelial hyperplasia in murine models, iv. IL-33 infusion-induced epithelial metaplasia and increased size and thickness of the biliary ducts, v. in susceptible mice, IL-33 triggering of ILC2s developed cholangiocarcinoma and liver metastasis [116].

### 4.2. Antitumor Activity

Despite the well-documented protumor role of type 2 immune responses, some reports provide evidence of antitumor activity of ILC2s, at least in certain types of cancer. These observations are in agreement with some data showing that type 2 shifting is associated with good prognosis in some tumors, such as some types of breast cancers, follicular lymphoma and Hodgkin’s lymphoma [117]. New reports indicate that, in different tissue, plastic ILC2s may selectively express distinct functional cytokine receptors for cell adaptation to local cues. Many ILC2s receptors for cytokines belonging to IL-1 family are encoded by genes colocalized in a very short genomic sequence whose regulation and expression are very specific to the cell types and/or tissue [118]. In response to the different tissue signals, ILC2s can follow some paths amplifying antitumor type 1-oriented response as the following: i. the recruitment and activation of eosinophils which may induce chemokines (CXCL9, CXCL10, etc) and cytokines (IFN-γ) production promoting the recruitment and M1 conversion of macrophages; ii. the production of IL-9 from ILC2s which contributes to M1 shift; iii. the directly activation of tumor-specific CD8+ T cells by ILC2s (exploiting their APC activity) which, in turn, produce IFN-γ, further contributing to amplify type 1 response, including ILC1s conversion from ILC2s (Figure 1B). Below we will analyze the main antitumor mechanisms of ILC2s.

*Eosinophil recruitment in TME.* Eosinophils can mediate antitumor activity via direct and indirect mechanisms [107]. ILC2- (or Th2) derived IL-5 enhances eosinophil survival and activation via IL-5Rα. Other cytokines such as IL-33 enhances the expression of LFA1 and CD11b on eosinophils and ICAM1 on tumor cells facilitating their interaction [119]. In response to several stimuli such as IL-5, IL-33, CCL11, IFNγ and TNFα, eosinophils secrete cytotoxic proteins (MBP, ECP, EDN and granzymes), which can induce apoptosis of tumor cells [119,120]. Importantly, eosinophil-derived cytokines such as IL-12 and IL-10, can also decrease metastasis by enhancing E-cadherin expression on tumor cells and improving their adhesion [121]. In addition, eosinophils express some NK-activating receptors (2B4, NKG2D and LY49), which may exert cytolytic activity to tumor cells by interacting with several ligands (MHC class I, MIC and CD48) on their surface [122]. Indirectly, eosinophils promote antitumor activity by releasing IFNγ, which, in an autocrine fashion, allows the production of pro-Th1-related chemokines (CXCL9, CXCL10, CXCL11) inducing CD8+T cells homing and cytotoxicity. Notably, the same pro-Th1 chemokines display an antiangiogenetic effect [123]. Finally, cytokine (IFNγ, TNF-α)-activated eosinophils polarize macrophages to an antitumorigenic M1 profile [107] or shift ILC2s to ILC1s with cytolytic activity.

In vivo studies on ILC2s-driven eosinophils in cancer indicate that ILC2-produced IL-5 promotes blood and tissue eosinophilia that positively correlates with reduced tumor growth. This activity was significantly upregulated in IL-5tg mice showing a massive eosinophils infiltration within and surrounding tumors [124]. Conversely, tumor growth was markedly increased in *Ccl11–/–* mice, *Il5–/–;Ccl11–/–* and eosinophil-deficient mice [125]. Administration of anti-IL-5Rα monoclonal antibodies (mAbs) resulted in decreased eosinophil levels and increased tumor cell growth [122]. On the whole these data suggest that, under specific settings, eosinophils can carry out antitumor activity.

*CXCR2+ tumor cells apoptosis*. It was reported that local IL-33 production could inhibit the tumor growth of a lymphoma cell line, by recruiting eosinophils into tumors through the overproduction of IL-5 from the IL-33-driven ILC2s [126]. Only mice that received wild type ILC2s, but not ILC2s from IL-33R-deficient mice, control the tumor growth [127]. In these models ILC2s produced CXCR2 ligands (CXCL1 and CXCL2) inducing apoptosis of CXCR2+ tumor cells [65]. It has been recently shown that some human tumor cells from PDAC can produce TSLP leading to increased survival and activation of ILC2s/IL-5/Eosinophil axis which in turn could impair tumor growth by a similar mechanism [76].

*Expansion of tumor-associated antigens- (TAA) specific CD8 T cells.* Another mechanism of antitumor activity of ILC2s is the expansion of CD8+ T cells specific for the TAA [128]. As previously described, it has been shown that ILC2s may exert APC activity due to their expression of MHC-Class I/II antigens and costimulatory molecule OX40L [45,58,64]. Thus, ILC2s can directly activate tumor-specific CD8+T cells, since are localized at interfollicular region surrounding B cell follicles of tumor-draining lymph nodes. By comparing the proportion and function of ILC2s infiltrating two different types of lung cancer (a primary tumor producing high IL-33 and a metastatic tumor producing low IL-33), ILC2s were reduced in metastatic- in comparison to primary tumor. However, when mice were grafted with the metastatic tumor cells engineered to produce IL-33, the proportion of ILC2s was higher and the tumor growth and metastases were reduced [128]. In vitro experiments of cocultures of metastatic prostate tumor cells from mice bearing ILC2s with OVA-pulsed mDCs and OVA-specific T cells, a higher killing capacity of the latter was observed, likely due to the increased expression of MHC class I on tumor cells [129].

Finally, as previously described, it has been recently reported that ILC2s infiltrating PDAC are able to activate tissue-specific tumor immunity. IL-33 was shown to activate tumor ILC2s and CD8+ T cells in orthotopic pancreatic (but not heterotopic skin) tumors to impair pancreas-specific tumor growth. This report identifies ILC2s as novel targets of a PD1 immunotherapy. Indeed in these tumors resting and activated infiltrating ILC2s expressed PD1, whose blockade results in ILC2s expansion with the improvement of tumor control and efficacy of therapy [87].

## 5. ILC2s as Therapeutic Targets in Tumors

Among ILCs, ILC2s are mostly considered a protumor subset, since the type 2 cytokines that they produce are associated with tumor growth and blocking antitumor immunity in TME [90,130,131]. However, many aspects still need to be elucidated to better understand the mechanisms behind ILC2 pro- and antitumor functions [131]. Among them, it is relevant to know whether ILC2s may act not only locally, but if, under certain conditions, they can migrate to distant sites to provide immune defense. Which are the conditions for homing and retention of ILC2s into TME of different tumors and if this happens for sites not containing resident ILC2s is matter of investigation.

Among alarmins, IL-33 has been extensively studied even though its role in pro- and antitumor immunity remains controversial. Several reports indicate that IL- 33 expression can be regulated during the progression of cancer and that signaling on IL-33R in TME may contribute to promoting antitumor responses or, in contrast, mediating tumor growth or metastasis. It has been suggested that these opposing effects may depend on the histology of malignant tissue, IL-33 local concentration [132], and the presence of different IL- 33 isoforms, as a result of post- translational processes or mRNA alternative splicing. The mutation level of IL-33 gene observed in human solid tumors may be associated with different isoforms of IL-33 [133]. Indeed the use of intratumoral injections of IL-33 promotes antitumor activity of ILC2s either through IL-5 and GM-CSF, which control tumor growth, through the recruitment of eosinophils or through the production of CXCR2 ligands with lytic activity on CXCR2+ tumor cells [126]. Exogenous IL-33 may stimulate the antitumor activity of CD8+T and NK cells, but it also expands intratumor ILC2s expressing the immunosuppressive ecto-enzyme CD73 [94,95,96]. In contrast, by using models of lung metastasis, IL-33-dependent ILC2s promote tumor burden and metastasis, mainly regulating type 1 activity of lung NK cells [133]. Therapeutic targeting of IL-33 or IL-5 reversed NK cell suppression and impaired cancer burden. In line with a protumor function of IL-33 in human colon–rectal carcinoma (CRC), the expression of the soluble form of IL-33R, was inversely associated with the malignant growth of CRC and metastasis [134].

Another possible target for the treatment of human APL is represented by the ILC2–MDSC axis, mostly driven by ILC2-produced IL-13: blocking any steps of this axis reversed the immunosuppression and significantly prolonged the survival time in humanized leukemic mice, while treating APL with retinoic acid reversed the increase in ILC2-induced MDSCs, as well as tumor- and ILC2-derived factors, in human patients with APL [32].

Other therapeutic targets related to ILC2s are the IL-4 and IL-13 receptors which are overexpressed in various kinds of tumors [100,135,136] and activated by their ligands which are largely produced by intratumor ILC2s. IL-4-IL-4R signaling is involved in CD133+ cancer stem cell survival and proliferation in CRC, which are therapy-resistant cells due to the autocrine production of IL-4 [137,138]. It has been shown that cancer initiating cells inhibited T cell proliferation in vitro via their membrane-associated IL-4 and IL-4R, while blocking IL-4-impaired tumor growth and stimulating antitumor T cell effectors [139]. As described, IL-4 production can regulate IL-10 synthesis by ILC2s through the upregulation of cMaf and Blimp-1 transcription factors. IL-10+ILC2s utilize both autocrine and paracrine signaling to suppress proinflammatory ILC2s effector functions, shifting from the fatty acid oxidation pathway usually utilized for proinflammatory activity [70,71]. The overexpression of IL-4Rα in tumor cells is usually associated with poor prognosis in some human cancer (bladder, head–neck carcinomas, etc.) [140,141], suggesting that IL-4Rα-targeted therapy would be appropriate for advanced/stage tumors. Among them a fusion protein composed of IL-4 and a truncated form of pseudomonas exotoxin has been carried out in phase I/II clinical trials of relapsed gliomas, leading to an overall higher survival in treated patients (NCT00014677 <https://clinicaltrials.gov /ct2/ show/ NCT00014677). Similarly, intratumor administration of a compound targeting IL-4R and bound to a cellular lytic peptide has been shown to inhibit pancreatic tumor, breast, head and neck and biliary tract carcinomas in vivo [142,143]. Lastly, IL-4R-targeted liposomal doxorubicin inhibited tumor growth more than chemotherapy in an orthotopic glioma xenograft model [144]. The IL-13Rα2 chain is also highly expressed by several tumor cells and binds with higher affinity to IL-13 than the other chain (IL-13Rα1), even when coupled to IL-4Rα. Due to its antiapoptotic and tumor-growing activity in IL-13Rα2-bearing cells, IL-13Rα2 has been proposed as a new oncogene [145]. Preclinical studies and clinical Phase I/II trials on glioblastoma (GB) patients have been performed, targeting this molecule [146]. The administration of surface-modified viruses expressing IL-13 was successful for prolonging survival in glioma mouse models [147]. Another approach was the anti-IL-13Rα2 mAb, or its FV fragment, coupled to cytotoxic drugs or radionuclide, which is always applied to GB [148]. Finally, a phase II clinical trial, designed with DNA vaccine by pulsing DC with GB antigens (or their peptides) plus IL-13Rα2 peptides showed a significant increase in the free-survival period [149]. Preclinical studies using intratumor injection of first and second generation CAR-T cells targeting IL-13Rα2 resulted in the regression of established human GB orthotopic xenograft mainly with IL-13Rα2-bearing tumors and secretion of type 1 cytokines [150]. Such studies provide a rationale for therapeutically targeting both receptors, which is also to be considered complementary to other antitumor approaches. The anti-IL-4Rα mAb has recently been approved by regulatory agencies for the treatment of some allergic disorders as well as other mAbs targeting molecules/receptors of type 2 inflammation [59], which could show promising results at least in some tumors.

Among ILC2s-derived cytokines, a promising therapeutic target is AREG, which induces TGF-β to converts NK cells into nonprotective helper ILC1s [151]. In addition, AREG (which is the ligand of EGFR) may induce an immune-suppressive TME by amplifying Tregs [152]. EGFR antagonists, which are in clinical use against epithelial-derived metastatic cancers or other AREG inhibitors, could be considered a new strategic approach at least in some tumors.

Finally, some molecules interfering with ILC2s proliferation may be of potential interest in cancer. It has been shown in a melanoma model that tumor-derived lactate attenuates the function and survival of ILC2s. Melanomas with reduced lactate production show delayed growth and an increased number of ILC2s compared with control tumors. A negative correlation has been shown between the expression of lactate and markers associated with IL-33/ILC2s/Eosinophil axis. The lactate production by melanoma cells may be considered a new therapeutic target to block tumor growth [153]. Recently, it has been demonstrated that Vit A deficiency induces the ILC2s infiltration of lung carcinoma with improved outcome, at least in the murine model [154]. If confirmed in the human setting, this discovery may provide a simple but effective strategy for the treatment of lung cancer.

## 6. Conclusions

Even though the number of studies focusing on the role of ILC2s in controlling tumor growth has increased considerably in the last decade, at present, a clear definition of these cells in TME of different tumors is not available [155,156,157]. In the future, the establishment of novel mouse models constitutively or transiently lacking ILC2s will be crucial to better define the role of these cells in conditioning tumor immunity. The use of humanized mice to establish patient-derived xenograft (PDX) models will also be useful to clarify this point.

Due to the high plasticity of these cells, which can easily adapt to the environment to which they are exposed the study of TME-infiltrating ILC2s in each patient may provide information on their pro- and/or antitumor roles. Another crucial point to establish is the contribution of ILC2s-recruited eosinophils and tumor cells themselves, which produce molecules affecting nILC2s and iILC2s in different tumor settings. There is clear evidence that, at least in some tumors, the immune response depends on the type of TME, which may change due to the contribution of three main components (tumor cell, ILC2 and eosinophil), which affect each other in promoting the recruitment of protumor suppressive cells or antitumor effectors in different tissues (Figure 1).

A better understanding of mechanisms of ILC2-mediated regulation of antitumor T cells will help to develop new strategies for immunotherapeutic interventions and improve existing treatment options. The anticytokine-based immunotherapies seem to be the most effective for changing ILC2s functions. However, type 2 cytokines are a challenging and nonspecific target due to the diversity of cell types producing these molecules and their multiple roles in different pathophysiological conditions. By exploiting the flexibility of ILC2s and Th2 cells toward a protective type 1 profile, a promising approach could be the development of adjuvant compounds of TLR ligands conjugated to molecules selectively entering intratumor macrophages and capable of redirecting the type 2-oriented TME. Another strategy may involve the disruption of transcription factor signatures useful for the ILC2 biology, or the modulation of their metabolic programs. In conclusion, there is an urgent need to perform further studies to investigate the expression of these cells interacting in the TME with other subsets of innate and adaptive immunity, in order to define their precise role in different human tumors and to establish effective antitumor therapies.

## Figures and Tables

**Figure 1 cancers-12-03452-f001:**
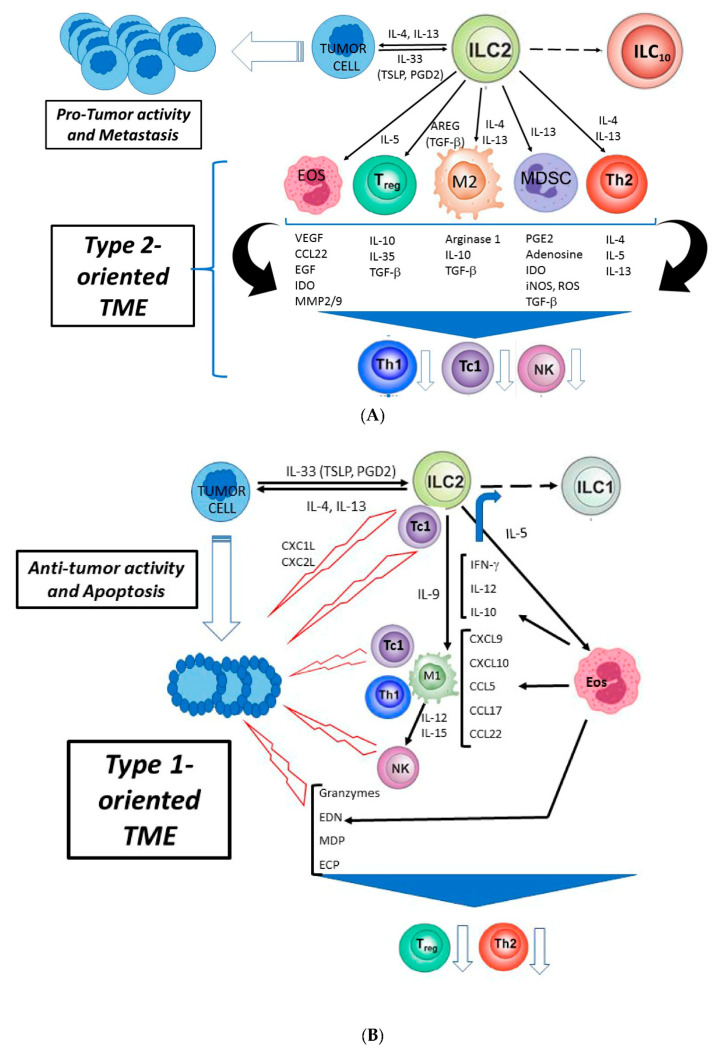
Opposite effects of group 2 innate lymphoid cells (ILC2s) on tumor burden. (**A**). Protumor activity of ILC2s: ILC2s interact with TME- and tumor cells by recruiting and amplifying cells of type 2 inflammation which favor tumor growth and metastasis. (**B**). Antitumor activity of ILC2s: ILC2s interact with TME- and tumor cells by recruiting and activating eosinophils, macrophages with M1 profile and other cells of type 1 response (CD8+T cells) and molecules (CXCL1L/CXCL2L) promoting the lysis/apoptosis of tumor cells. These opposite effects of ILC2s on tumor burden depend on the distribution, differentiation and plasticity of ILC2s in different tissues, the tumor histology, the TME signals activating ILC2s, the prompt and variable expression of signals (surface molecules and cytokines) activating the effector or regulatory cells of innate and adaptive immunity. AREG: amphiregulin, Eos: eosinophils, ILCs: innate lymphoid cells; ILC_10_: IL-10-producing ILCs; MDSC: myeloid-derived suppressor cells, M1, M2: functional profiles of tumor-associated macrophages, Tc: cytotoxic CD8+T cells; Th: T helper cells; TME: tumor microenvironment; Treg: regulatory T cell.

**Table 1 cancers-12-03452-t001:** Group 2 innate lymphoid cells (ILC2s) phenotype, distribution and pro- and antitumor function in human.

Tumor Types	Phenotype	Sites	Pro/Anti-Tumor	Associated Cells	Serum Cytokines	Functions	Ref
AML	ILC2s (also ILC1/ILC3)	PBMC	Protumor			Reduction	[79]
APL	Lin-CD127+NKp46- CRTH2+c-kit−/+	PBMC	Protumor	MDSC	IL-13	Increased	[32]
Breast cancer	Lin-CD127+CD56- CRTH2+CD117−/+	Tissue	Protumor	MDSC	IL-13	Increased Tregs and PD1+CTLA4+KLRG2+ILC2s (vs circulating ILC2s)	[32,80]
Lung cancer	Lin-ICOS+IL−17RB+	PBMC	Protumor	MDSC	IL-5, IL-13, IL-33, Arg1	Increased Reduction (vs blood)	[81] [82,83]
Gastric cancer	Lin-CD127+CRTH2+ CD161+ Lin-ICOS+IL−17RB+	Tissue PBMC	Protumor	MDSC, M2 type Macrophages	IL-4, IL-5, IL-25, IL33	Increased	[84]
Colorectal cancer	Lin-CD127+CRTH2+	Tissue	Protumor			Increased among Non-NK ILCs	[83,85]
Bladder cancer	Lin-CD127+CRTH2+ CD117−/+	Urine during BCG treatment	Protumor	MDSC	IL-13	T/MDSC ratio predictive of relapse	[86]
Prostate cancer	Lin-CD127+CRTH2+ CD117−/+	PBMC	Protumor	NKp30+ILC2 MDSC		ILC2s related to disease severity	[32]
Pancreatic Ductal Cancer	Lin-CD127+ST2+	Tissue	Antitumor	Activated CD8^+^T cells	IL-33	Increased in long-term survivors	[87]

AML: Acute Myeloid Leukemia; APL: Acute Promyelocytic Leukemia; Arg1: Arginase 1; BCG: Bacillus Calmette-Guerin; MDSC: myeloid-derived suppressor cells; PBMC: peripheral blood mononuclear cells.

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
