# Peer review of "Group 2 Innate Lymphoid Cells: A Double-Edged Sword in Cancer?"

_cancers, 2020, doi:10.3390/cancers12113452_

Round 1

Reviewer 1 Report

The authors of this review investigate the role of ILC2s in cancer. They comprehensively summarize the general knowledge of ILC2 biology, as well as what is currently known in the field regarding both pro- and anti-tumor activity from ILC2s. The review largely is well inclusive, but a number of supplemental revisions are required to strengthen the review.  

Major Revisions:

  1. There is an incredible number of typos and grammatical errors throughout the manuscript, including in the title as seen above. Please heavily revise. Please also make subheadings more clear, either through bold font or a different font size.
  2. In section 2, the authors briefly discuss regulatory molecules expressed by ILC2s, including ICOS/ICOS-L and ICAM/LFA-1. Please also discuss inhibitory molecules, including PD-1. Their expression is especially important to mention in relationship with current cancer therapeutics on the market.
  3. The manuscript seems to focus a little too much on the role of ILC2s in other diseases at the beginning of the paper. This can be shorted as it does not pertain to the main subject of the review.
  4. Overall, the paper requires more figures to conceptualize the role of ILC2s in specific TMEs as the role of ILC2s can differ depending on the cancer type.
  5. The review seems to be missing a key part of ILC2s which is their plasticity pertaining to ILC1s which would be able to fight the TME. Promotion of these populations would promote tumor immunity. The authors briefly mention ILC and ILC2 plasticity in section 2 and in the conclusion. Are there any studies focusing on ILC2 plasticity in the TME? Largely the pro- and anti- tumor properties reported from ILC2s in this review focus on cytokines IL-4, IL-5, and IL-13, the conventional cytokines secreted by ILC2s. But as the authors mention, ILC2s can be induced to produce many different cytokines based on their environment – in this case, the TME. Are there any reports investigating plasticity in this environment?
  6. Figure 1. It is not explicitly clear on how IL-5 and IL-13 contribute to a type 1 TME as both cytokines are implicated in the promotion of M2 macrophages in a variety of context leading to secretion of IL-10 and immune suppression.
  7. It will be important to highlight the homing and retention of ILC2s in different tumor types as it is relevant as some tumors, purely based on location, do not contain resident ILC2 populations.
  8. Line 182. Paragraph highlighting the existence of IL-10+ ILC2s fails to relate that to cancer progression and the generation of these cells in the TME.
  9. Similar to the above observation, papers on IL-10+ ILC2s recently came out (PMID: 31699824, PMID: 32905799 linking IL-4 stimulation with an anti-inflammatory phenotype for ILC2s. These papers should be cited and discussed, especially in the paragraph starting in 388 that discusses how IL-4R-IL-4 signaling correlates with a poor prognosis in a variety of human cancers.
  10. For the heading paragraph of section 4, are there any IL-25 enriched tumors?
  11. Line 221 – change the word “improved”. As it stands, it sounds like the tumor growth is reduced, not grown.
  12. It is essential that the authors create a chart listing the cancer types ILC2s have been reported to have pro- or anti-tumor activity, as well as the original manuscript source.

Author Response

The authors of this review investigate the role of ILC2s in cancer. They comprehensively summarize the general knowledge of ILC2 biology, as well as what is currently known in the field regarding both pro- and anti-tumor activity from ILC2s. The review largely is well inclusive, but a number of supplemental revisions are required to strengthen the review.  

 Major Revisions:

  1. There is an incredible number of typos and grammatical errors throughout the manuscript, including in the title as seen above. Please heavily revise. Please also make subheadings more clear, either through bold font or a different font size.

We required the support of MDPI English editing Service which revised the draft by correcting typos and grammatical errors. Subheadings have been made clearer by using a different font size.

  1. In section 2, the authors briefly discuss regulatory molecules expressed by ILC2s, including ICOS/ICOS-L and ICAM/LFA-1. Please also discuss inhibitory molecules, including PD-1. Their expression is especially important to mention in relationship with current cancer therapeutics on the market.

We agree with the reviewer on the importance of other regulatory molecules expressed by ILC2s, as PD1.  Due to their pro- or antitumor activity, however, the precise role of such a molecule on ILC2s is at present unknown: this is the reason why this point has not completely covered in the original version. Anyway, a new paragraph with related refs has been added to the new text pointing out the PD1 expression on ILCp and mature ILC2s, the pathophysiological function on ILC2s in some in vitro and in vivo models (including cancer), the pro- or antitumor effects of anti-PD1 inhibitors in different tumors (see lines 285-308)

  1. The manuscript seems to focus a little too much on the role of ILC2s in other diseases at the beginning of the paper. This can be shorted as it does not pertain to the main subject of the review.

We agree with the reviewer that the section on the role of ILC2s in other diseases are too much expanded. Even though some aspects of pathophysiology of ILC2s must be highlighted  before describing their role in TME, we shortened this section either by eliminating some parts or postposing some general concepts (applicable also to the ILC2s in tumors) in the subsequent sections examining the pro- and antitumor activity of ILC2s (see lines 229-238, 261-265, 268-274, 365-369)

  1. Overall, the paper requires more figures to conceptualize the role of ILC2s in specific TMEs as the role of ILC2s can differ depending on the cancer type.

We thank the reviewer for this suggestion which allows to expand and point out the role of ILC2s in specific TMEs. Accordingly, we have modified the Fig. 1 by splitting it into two panels analysing the major known mechanisms able to impair or improve tumor growth and metastasis. In addition, since the role of ILC2s can differ depending on the type of cancer, the new version of the paper has been enriched with the Table 1 reporting the pro- or antitumor activity of ILC2s in different human cancers.

  1. The review seems to be missing a key part of ILC2s which is their plasticity pertaining to ILC1s which would be able to fight the TME.

We thank the reviewer for this suggestion. In the previous version of the paper the shift of ILC2s to ILC1s has been described (see lines 122-124) in the inflamed gut of Crohn’s disease [48] and in the lung of severe COPD patients [49]. An important point on ILC1s modulated-cells concerns their acquired cytolytic activity or their possible origin from a cytotoxic ILC2s subset which could play a role in the TME of some tumors. This latter subset coexpressing CRTH2 and CD94 was never observed in normal tissues as tonsils. However, since CRTH2 is downregulated in ILC2s upon activation, there is the possibility that cytotoxic ILC2s are CRTH2 negative and are, thus, missed in ex vivo analysis of cells infiltrating TMEs. This latter concept has been added to the text (see lines 118-122, 333, 348-349) and reported in the Fig. 1b.

  1. Promotion of these populations would promote tumor immunity. The authors briefly mention ILC and ILC2 plasticity in section 2 and in the conclusion. Are there any studies focusing on ILC2 plasticity in the TME?

This is an interesting field of research. Poor and not conclusive data have been published on the plasticity of intratumoral ILCs. Some new findings indicate that plastic ILC2s in different tissue may selectively express distinct functional cytokine receptors for cell activation in response to local cues. In particular many of these receptors belonging to IL-1 family cytokines are encoded by genes co-localized in a very short genomic sequence and their regulation and expression are very specific to the cell types and/or tissue [see new ref. 118]. The concept of tissue adaptation of ILC2s has been reaffirmed in the text (see lines 324-327).

  1. Largely the pro- and anti- tumor properties reported from ILC2s in this review focus on cytokines IL-4, IL-5, and IL-13, the conventional cytokines secreted by ILC2s. But as the authors mention, ILC2s can be induced to produce many different cytokines based on their environment – in this case, the TME. Are there any reports investigating plasticity in this environment?

As previously stated, poor and not conclusive data are at present available on the plasticity of ILC2s inside the TME. This notion has been reported in different sections along the text and in the conclusions of the old and new versions (see lines 469-476)

  1. Figure 1. It is not explicitly clear on how IL-5 and IL-13 contribute to a type 1 TME as both cytokines are implicated in the promotion of M2 macrophages in a variety of context leading to secretion of IL-10 and immune suppression.

Figure 1 has been completely renewed. It has been divided into two parts reporting the main protumor and antitumor mechanisms of ILC2s. As stated by the reviewer, Fig 1 was not explicity clear on how ILC2s contribute to a type 1 response in TME response. Due to the different tissue signals ILC2s can follow some paths amplifying type 1-response, which, parallely, inhibits type 2-oriented microenvironment. The known paths are: the recruitment and activation of eosinophils which under some stimuli may induce the production of chemokines (CXCL9, CXCL10, etc) and IFN-g promoting the recruitment and M1 conversion of macrophages; ii. The production of IL-9 from ILC2s which contributes to M1 shift; iii. the directly activation of tumor-specific CD8+ T cells by ILC2s (exploiting their APC activity) which, in turn, produce IFN-g, further contributing to amplify type 1 response. These concepts have been reported in the text of the new version (see lines 327-334).  

  1. It will be important to highlight the homing and retention of ILC2s in different tumor types as it is relevant as some tumors, purely based on location, do not contain resident ILC2 populations

This is, indeed, an opened problem: ILC2s may act not only locally, but, under certain conditions, migrate to distant sites to provide immune defense. Which are the conditions for homing and retention of ILC2s into TME of different tumors and if this happens for sites which do not contain resident ILC2s is matter of investigation. Further studies on the trafficking of iILC2 into TME may provide the definition of novel targets for immunotherapy. This concept has been added in the section 5 of the new version (see lines 387-390).

  1. Line 182. Paragraph highlighting the existence of IL-10+ ILC2s fails to relate that to cancer progression and the generation of these cells in the TME.

We agree with the reviewer that the paper did not highlight the role of IL-10+ ILC2s in cancer progression and the potential signals generating these cells in TME. Unfortunately, the in vivo suppressive role of these cells has been reported in murine models of lung inflammation (COPD and asthma), but poorly in cancer. This concept has been repeated in the text (see lines 236-239; 280-284)

  1. Similar to the above observation, papers on IL-10+ ILC2s recently came out (PMID: 31699824, PMID: 32905799 linking IL-4 stimulation with an anti-inflammatory phenotype for ILC2s. These papers should be cited and discussed, especially in the paragraph starting in 388 that discusses how IL-4R-IL-4 signaling correlates with a poor prognosis in a variety of human cancers.

The suggested paper (PMID: 31699824 and PMID: 32905799) linking the IL-4 production with the suppressive activity of IL-10+ILC2s in murine models, has been quoted and discussed in the new version (refs. 70 and 71) (see lines 168-175; 419-422)

  1. For the heading paragraph of section 4, are there any IL-25 enriched tumors

Several primary tumors and tumor cell lines have been described to produce alarmins as IL-25 and TSLP (see new refs 77, 78). However, as described, only inflammatory ILC2s (iILC2s) are responsible to exogeneous IL-25, while IL-33 may trigger both nILC2s and iILC2s in association (or not) with IL-25. For this reason IL-25 is not considered a good therapeutical target. About TLSP it has been shown that such alarmin may improve tumor growth by triggering TSLP receptor expressed by the same tumor cells in an autocrine manner and independently from a type 2 inflammation. As required, these concepts have been added at the heading paragraphs of section 4 of the new version (see lines 184-190)  

  1. Line 221 – change the word “improved”. As it stands, it sounds like the tumor growth is reduced, not grown.

The word in line 221 of the old version has been changed into “increased”

  1. It is essential that the authors create a chart listing the cancer types ILC2s have been reported to have pro- or anti-tumor activity, as well as the original manuscript source.

A new table has been added in the new version of the manuscript listing the cancer type in which ILC2s have been reported to exert pro- or antitumor activity as well as quoting the original manuscripts (see line 193).

Reviewer 2 Report

This manuscript, written by Dr. Enrico Maggi et al., with the title of “Group 2 Innate Lymphoid Cells: A Ddouble-Edged Sword in Cancer?” is a review that focuses on Group 2 innate lymphoid cells (ILC2s).

Innate lymphoid cells (ILCs) are a recently identified immune cell type that is found in almost every tissue but is specifically enriched at mucosal surfaces, where they are thought to play an important role in maintaining epithelial barrier integrity and regulating immune responses. Several subsets of ILCs have been characterized based on differences in their phenotypes and functional properties. These include group 1 ILCs, which consist of natural killer cells (NK cells) and ILC1 cells, group 2 ILCs (ILC2s), and a heterogeneous population of group 3 ILCs (ILC3s), which includes (NCR+) ILC3s, (NCR-) ILC3s, and fetal lymphoid tissue-inducer cells (LTi cells).

ILC2s, which express the transcription factor, GATA-3, and produce Th2-like cytokines in response to large extracellular pathogens have been suggested to be the innate equivalent of Th2 cells. Exaggerated ILC2 immune responses are associated with atopic diseases such as asthma, chronic rhinosinusitis, and atopic dermatitis. In humans, ILC2s are commonly identified as Lin-CD127+CD45highCD161+CRTH-2+ cells that also express ST2 and IL-17 RB.

This review focuses on ILC2s and in 10 pages makes a very thorough revision of the literature on this topic. The authors divided the text in several parts: (1) Introduction, (2) Main fatures of ILC2x, (3) Enhancing and regulatory functions, (4) Pro- and anti- tumor activities, (5) Pro- tumor, (6) Anti-tumor, (7) Therapeutic targets in tumors, and (8) Conclusions.

This review contains only one figure in which the authors explain the pro- and anti-tumor activities of the ILC2s. This review doesn’t have any table.

The text it is well written, and it is easy to read and follow, although there are parts that contain a lot of information and will require concentration from the reader, specially if it is not his specialty.

Before publishing this review manuscript, the authors could address the following minor comments:

1- The title is “Group 2 Innate Lymphoid Cells: A Ddouble-Edged Sword in Cancer?”. I think that there is a typographic error and should be “A Double-Edged Sword”.

2- The authors could add some tables that try to summarize the most relevant sections of the text. Since the text is dense in some parts, a table could simplify the information.

3- Figure 1 is important because it deals with the relationship of ILC2s with cancer, either towards a pro-tumoral immune microenvironment with “type-2” polarization or an anti-tumoral “type-1”. This figure legend could be expanded a little more, with more information of the effect of each of the TME cells in the tumoral cells or other components of the microenvironment. There is a review about ILC2s (https://doi.org/10.3389/fimmu.2019.02801) that may provide some additional ideas that could be incorporated in this review.

Author Response

This manuscript, written by Dr. Enrico Maggi et al., with the title of “Group 2 Innate Lymphoid Cells: A Ddouble-Edged Sword in Cancer?” is a review that focuses on Group 2 innate lymphoid cells (ILC2s).

Innate lymphoid cells (ILCs) are a recently identified immune cell type that is found in almost every tissue but is specifically enriched at mucosal surfaces, where they are thought to play an important role in maintaining epithelial barrier integrity and regulating immune responses. Several subsets of ILCs have been characterized based on differences in their phenotypes and functional properties. These include group 1 ILCs, which consist of natural killer cells (NK cells) and ILC1 cells, group 2 ILCs (ILC2s), and a heterogeneous population of group 3 ILCs (ILC3s), which includes (NCR+) ILC3s, (NCR-) ILC3s, and fetal lymphoid tissue-inducer cells (LTi cells).

ILC2s, which express the transcription factor, GATA-3, and produce Th2-like cytokines in response to large extracellular pathogens have been suggested to be the innate equivalent of Th2 cells. Exaggerated ILC2 immune responses are associated with atopic diseases such as asthma, chronic rhinosinusitis, and atopic dermatitis. In humans, ILC2s are commonly identified as Lin-CD127+CD45highCD161+CRTH-2+ cells that also express ST2 and IL-17 RB.

This review focuses on ILC2s and in 10 pages makes a very thorough revision of the literature on this topic. The authors divided the text in several parts: (1) Introduction, (2) Main fatures of ILC2x, (3) Enhancing and regulatory functions, (4) Pro- and anti- tumor activities, (5) Pro- tumor, (6) Anti-tumor, (7) Therapeutic targets in tumors, and (8) Conclusions.

This review contains only one figure in which the authors explain the pro- and anti-tumor activities of the ILC2s. This review doesn’t have any table.

The figure has been completely changed and divided into two panels describing the main mechanisms promoting the pro- or the antitumor activities of ILC2s. A new table has been also added which summarizes the different human tumors in which the distribution, phenotype and mechanisms leading to pro- or antitumor activity of ILC2s have been shown (see line 194)

The text it is well written, and it is easy to read and follow, although there are parts that contain a lot of information and will require concentration from the reader, specially if it is not his specialty.

Before publishing this review manuscript, the authors could address the following minor comments:

  • The title is “Group 2 Innate Lymphoid Cells: A Ddouble-Edged Sword in Cancer?”. I think that there is a typographic error and should be “A Double-Edged Sword”.

The type error has been corrected: we also required the support of MDPI English editing Service which heavily revised the draft by correcting typos and grammatical errors.

  • The authors could add some tables that try to summarize the most relevant sections of the text. Since the text is dense in some parts, a table could simplify the information.

Table 1 has been added to the paper: it summarizes the distribution, phenotype and mechanisms leading to pro- or antitumor activities of ILC2s shown in different human tumors

  • Figure 1 is important because it deals with the relationship of ILC2s with cancer, either towards a pro-tumoral immune microenvironment with “type-2” polarization or an anti-tumoral “type-1”. This figure legend could be expanded a little more, with more information of the effect of each of the TME cells in the tumoral cells or other components of the microenvironment. There is a review about ILC2s (https://doi.org/10.3389/fimmu.2019.02801) that may provide some additional ideas that could be incorporated in this review.

According to the reviewer’s suggestion and relating to the paper (https://doi.org/10.3389/fimmu.2019.02801) already quoted in the old version [142], the figure 1 has been completely changed: in the new version it is divided into two parts describing the main mechanisms promoting the pro- or the antitumor activities of ILC2s.

Round 2

Reviewer 1 Report

The authors performed an excellent job and edited the review sufficiently. My recommendation is to publish.